# Application of Selected Biomaterials and Stem Cells in the Regeneration of Hard Dental Tissue in Paediatric Dentistry—Based on the Current Literature

**DOI:** 10.3390/nano11123374

**Published:** 2021-12-13

**Authors:** Alina Wrzyszcz-Kowalczyk, Maciej Dobrzynski, Iwona Grzesiak-Gasek, Wojciech Zakrzewski, Monika Mysiak-Debska, Patrycja Nowak, Malgorzata Zimolag, Rafal J. Wiglusz

**Affiliations:** 1Department of Pediatric Dentistry and Preclinical Dentistry, Wroclaw Medical University, Krakowska 26, 50-425 Wroclaw, Poland; alina.wrzyszcz-kowalczyk@umw.edu.pl (A.W.-K.); iwona.grzesiak-gasek@umw.edu.pl (I.G.-G.); monika.mysiak-debska@umw.edu.pl (M.M.-D.); patrycja.nowak@student.umw.edu.pl (P.N.); malgorzata.zimolag@student.umw.edu.pl (M.Z.); 2Pre-Clinical Research Centre, Wroclaw Medical University, Bujwida 44, 50-345 Wroclaw, Poland; 3Institute of Low Temperature and Structure Research, Polish Academy of Sciences, Okolna 2, 50-422 Wroclaw, Poland; r.wiglusz@intibs.pl

**Keywords:** paedodontics, remineralization, nanohydroxyapatite, stem cells, fissure sealants, pulp capping, restorative materials, ART technique

## Abstract

Currently, the development of the use of biomaterials and their application in medicine is causing rapid changes in the fields of regenerative dentistry. Each year, new research studies allow for the discovery of additional possibilities of dental tissue restoration. The structure and functions of teeth are complex. They consist of several diverse tissues that need to act together to ensure the tooth’s function and durability. The integrity of a tooth’s enamel, dentin, cementum, and pulp tissue allows for successful mastication. Biomaterials that are needed in dentistry must withstand excessive loading forces, be biocompatible with the hosts’ tissues, and stable in the oral cavity environment. Moreover, each tooth’s tissue, as well as aesthetic qualities in most cases, should closely resemble the natural dental tissues. This is why tissue regeneration in dentistry is such a challenge. This scientific research focuses on paediatric dentistry, its classification of caries, and the use of biomaterials in rebuilding hard dental tissues. There are several methods described in the study, including classical conservative methods such as caries infiltration or stainless-steel crowns. Several clinical cases are present, allowing a reader to better understand the described methods. Although the biomaterials mentioned in this work are artificial, there is currently ongoing research regarding clinical stem cell applications, which have a high potential for becoming one of the most common techniques of lost dental-tissue regeneration in the near future. The current state of stem cell development is mentioned, as well as the various methods of its possible application in dentistry.

## 1. Introduction

Currently, tooth loss is a common and alarming problem that presents a concern for society [1] and for the general health of patients [2,3]. The resulting reduced chewing ability primarily affects the quality of the health and well-being of the patient. Patients in the developmental age visit the dentist’s office most often for carious lesions of their primary teeth, which are the cause of pain and chewing disorders, consequences of untreated caries in the form of pulp inflammation, and the cause of developmental disorders of tooth mineralization and post-traumatic damage to hard dental tissues. Caries is reported as the most common disease worldwide [4], and together with genetic disorders and periodontal diseases, it is a major cause of tooth loss.

The tooth is a complex organ, consisting of a mix of hard, mineralized tissues such as enamel and dentin, as well as soft tissues, including dental pulp [5]. It is important to emphasize that the outer layer of enamel cannot be biologically repaired except through the process of remineralization. Enamel tissue can be reinforced and remineralized using fluoride ions. As Volponi et al. [6] pointed out, newly formed hydroxyapatite often lacks the mechanical properties and structure of natural enamel, which makes recreating the hierarchical structure of the natural enamel the greatest challenge during the whole process.

There have been already a several attempts at whole-tooth regeneration in history. For instance, Young et al. [7] were able to regenerate the first-ever tooth, together with diverse tissues such as enamel and dentin, with the use of porcine third molar tooth buds. The regenerated tissues still had improper features, however.

Enamel, being the outermost dental layer and the hardest tissue in the human body, is subject to extensive mastication, pH changes, as well as caries [8]. Regeneration is not needed when the enamel is affected early without a cavitation. Then, it requires only remineralisation by the incorporation of amelogenins and the use of biomimetic routes [9]. The real advantage occurs when there is a need for enamel regeneration, as it is challenging to guide the regeneration process in order to arrange HAp crystals and prisms in a morphology found in natural teeth, requiring several steps to achieve enamel tissue from ameloblasts undergoing odontoblasts differentiation [10].

As for dentin, it has been successfully generated indirectly as a result of odontoblast-like cell formation, using postnatal mesenchymal cells such as dental papilla stem cells [11] or bone marrow mesenchymal stem cells [12].

The exceptional morphology and histology of dental pulp makes it by far the most challenging tissue for dental regeneration. Its access is limited due to its enclosure inside the tooth by dentin. Although it is well-innervated and vascularized, it lacks collateral circulations, meaning the activity of the immune system in infected cells’ eradication is limited [13].

In dentistry, there are numerous methods for the restoration of lost hard dental tissue. The examples discussed in this scientific work are not final, due to the constant and rapid development of biomaterials in medicine, especially in dentistry. Current dental treatments in cases of carious lesions are based on conservative approaches using inorganic materials. Some of the procedures focus on restoring lost biological tissues with artificial ones. However, such methods as those that use stem cells have promising potential for the regeneration of biological tissues after the removal of carious lesions. It has to be highlighted that these methods are still under ongoing scientific research, but their unique potential can be confirmed by numerous scientific research papers [14,15,16]. The progress of tissue engineering applications in medicine still requires extensive translation from the laboratory to the dental office. As Zafar et al. explains, it is a complicated process requiring advanced logistics [17]. The topic concerning future dental tissue regeneration is tempting, yet it needs extensive collaboration between dentists, biotechnologists, and biomaterials scientists. The approach of regenerative dentistry is based on understanding both the biological processes of healing and the different stages of tooth development.

This study aims to review the literature and present the various methods of hard dental-tissue reconstruction and regeneration in paediatric patients.

## 2. Materials and Methods

There is a plethora of methods and protocols for hard dental-tissue regeneration found in the literature. The authors have conducted a scientific search for hard dental-tissue regeneration in paediatric patients, presented in Figure 1. Based on a review of scientific publications on nanomaterials, the majority of searches were conducted in PubMed. The primary database contained 16,465 results. Later, additional filters were used to narrow the search results, including publication date, language, free text or full text, species, and age of the patients. Finally, 209 results were evaluated, and eventually, 144 results were chosen and compared. During the scientific search, Boolean logic was used [18].

## 3. Methods of Treatment and Classification of Dental Caries

The American Dental Association (ADA) and the American Academy of Pediatric Dentistry (AADP) have identified early childhood caries (ECC), defined as the presence of one or more decayed primary teeth removed or filled because of caries in children 71 months of age or younger, i.e., before the age of 6 [19]. Examples of ECC cases are shown in Figure 2. The conducted research proves that, in primary dentition, the progression of lesions limited to the enamel is, on average, faster than in permanent teeth [20]. On the other hand, changes within the dentin compared to enamel even more dynamically, both in primary and permanent dentition [21,22].

Nutritional deficits and oral tissue infections eventually lead to an imbalance in the natural oral flora, increasing the risk of health deterioration, as well as of a poorer quality of life [23,24]. In older children, we also observe a high susceptibility to caries in young, immature dentition. In this case, apart from the typical aetiological factors of caries disease, there are factors that determine the severity of changes in the anatomical features of young teeth, such as deep, difficult-to-clean grooves, and poorly mineralized hard tissues of the tooth.

The methods of caries treatment include non-invasive (non-interventional) methods, aimed at stopping the development of caries, and invasive (interventional) methods [25], involving the mechanical (surgical) removal of damaged tissues and their reconstruction. The most important diagnostic symptom of superficial caries is the lack of smoothness of the enamel surface, which is the most highly mineralized tissue of the human body [26]. A total of 90% of the inorganic matter of properly formed enamel are hydroxyapatite crystals. The remaining part is formed by an inorganic amorphous substance consisting of calcium, magnesium, and fluorine compounds [27].

### 3.1. White Spot Lesion (WSL) and Remineralization

Patients with early caries lesions, manifested by the lack of enamel gloss and chalky white or brown discolorations, represent a group with medical needs that primarily require the use of caries-prevention methods based on inhibiting the growth of cariogenic bacteria and remineralizing hard-tissue lesions of the teeth [28]. The remineralization of hard tooth tissues is a process in which previously lost minerals (calcium and phosphate ions) return to the tooth structure because of redeposition [29]. This process only occurs in the case of an undamaged surface layer characterized by subsurface demineralization.

The WSLs resulting from demineralization are reversible when the right amount of calcium and phosphate ions is provided [30]. In the group of compounds accelerating remineralization, fluoride compounds that inhibit demineralization, stabilize saliva pH at the physiological level, increase the absorption of calcium and phosphate ions and the formation of fluorhydroxyapatite [31], as well as the deposition of calcium fluoride, have found practical applications. As Figure 3 shows, and as other journal articles confirm, WSLs occur most commonly in the cervical area, as well as the incisal/occlusal surfaces of teeth [32,33]. Fluoride, at low concentrations, integrates with enamel prisms to form fluorhydroxyapatite or fluorapatite. However, in the case of high concentrations, it forms grains of calcium fluoride microcrystals containing phosphates [34]. These play the role of a stable reservoir of fluorine, enabling the precipitation of precipitates on the surface of the enamel, which constitute an ion-exchange barrier with subsurface enamel zones [35,36,37,38]. Among the treatments that inhibit the initial lesions of caries, remineralization is recommended with the use of fluoride varnishes applied at 1-, 3- or 6-month intervals, depending on the level of risk of caries [39]. Professional fluoride varnishes contain between 1000 and 22,600 ppm of F (Duraphat, Colgate, New York, NY, USA). When using them, observe the appropriate—safe—dose, and instruct the patient to refrain from eating for at least 45 min (and preferably for 4 to 6 h) after the procedure, as well as to not use other fluoride agents within 24 h of the application. Gels and foams with 5000–12,500 ppm of F (Elmex, Hamburg, Germany) should be applied on standard spoons (4 min); observe the appropriate dose of the preparation, tilt the patient’s head forward, use a saliva ejector, and advise the patient to spit out the saliva for 30 s after application. Compounds favouring remineralization are inorganic fluoride compounds, such as sodium fluoride (Fluor Protector, Ivoclar Vivadent, Schaan, Liechtenstein), as well as organic fluorine compounds—fluoramines—and their effectiveness is similar [40,41].

### 3.2. Nanohydroxyapatite and Remineralization

Bioactive (biologically compatible) materials are preparations that do not damage living cells in both in vitro and in vivo conditions, and that have minimal impact on the immune system [42]. In dental treatment, this type of preparation, showing biocompatibility based on direct integration with dentin at the molecular level, is tolerated by pulp cells and stimulates odontoblasts to produce secondary and reparative dentin, showing an odontotropic effect [43,44,45,46].

A bioactive and biocompatible alternative to the action of fluoride preparations is hydroxyapatite, which permanently bonds with enamel apatite, forming crystals with a regenerating effect, enabling the regeneration of microdamage in hard dental tissue.

Nanoparticles have unique properties that are influenced by their quantum size [47], which distinguishes them from materials with a larger particle scale. Examples of such characteristics may be wettability, charge, porosity, altered electronic structure, lower contact angle, increased surface area, number of surface atoms, or an increased surface area, which greatly influences the interaction with other materials [48]. Consequently, the addition of hydroxyapatite nanoparticles to the polymer matrix causes an elevation of the mechanical strength of the biomaterial.

One of the characteristics of nanohydroxyapatite particles with a size between 50 and 1000 nm is their similarity to the mineral structure of the tooth, with the chemical formula Ca_10_(PO_4_)_6_(OH)_2_ [49]. Hydroxyapatite contains calcium and phosphorus at a ratio of 1:67 [50]. It is the basic material present in the bones and teeth [51]. Enamel is the hardest dental tissue, containing relatively large nanohydroxy and fluorapatite crystals that are 25 nm thick, 40–120 nm wide, and 1000 nm long.

Nanohydroxyapatite paste first appeared on the European market in 2006 [52]. In a study published in 2018, comparing the effectiveness of fluoride and nanohydroxyapatite for remineralizing initial carious lesions in enamel, it was shown that, despite the lack of significant differences in the remineralization process between the tested compounds, the nHAp paste showed a promising long-term protective effect in terms of surface deposition, and also showed the effect of a smoother enamel surface after its application, compared to fluoride varnish.

The authors of the study pointed out that nanohydroxyapatite paste may be recommended as an alternative remineralizing agent, especially beneficial for children, pregnant women, and patients at risk of developing fluorosis [53]. Despite positive reviews of oral care formulations containing nanohydroxyapatite, regulatory concerns regarding the safety of their use have appeared, having been raised by the European Commission’s Scientific Committee on Consumer Safety (SCCS). Studies conducted in response to these concerns confirmed that nanohydroxyapatite is cytocompatible, and that it does not alter the normal behaviour of cells and is therefore safe for use in oral care products [54].

Nanohydroxyapatite, due to its biocompatibility [52], bioactivity [55], stability, and non-toxicity, is a promising material [56], whether in orthopaedics, maxillofacial surgery, or conservative dentistry [57]. In dental treatment, HAp can be incorporated into many products, including fillings, cements, mouthwashes, and gels [58]. Examples of agents initiating the remineralization of damage to hard dental tissues are calcium phosphate preparations containing nanoparticles of amorphous calcium phosphate (ACP), combined with casein phosphopeptide (CCP) at a ratio of 6 CPP molecules, combined with 144 particles of calcium and 96 phosphates—or Recaldent systems (e.g., Tooth Mousse preparations, MI fluoride toothpaste). The group of similarly acting preparations is represented by products based on hydroxyapatite in a wide range—for both individual and professional use: ApaCare (paste, gel, rinse), containing nanohydroxyapatite that is chemically almost identical to enamel hydroxyapatites marked as “Medical Hydroxyapatite” (to distinguish it from the abrasive agent).

The prophylactic effect of nanohydroxyapatite is based on three mechanisms: forming a protective layer of “liquid enamel” on the teeth, sealing damage to the enamel with fine nanohydroxyapatite crystals with a particle size of 20–80 nanometres, and accelerating the remineralization [59] of hard tooth tissues through the supplied phosphate and calcium ions, as can be seen in Figure 4. 

Nanohydroxyapatite in toothpaste has a high ability to bind to proteins, and also with components of dental plaque and bacteria, which is due to the size of the nanoparticles, increasing the surface to which proteins can attach, and at the same time, acting as a filler that penetrates porosity and micro-cavities on the surface of the enamel [54].

A new generation of oral care products contains nHAp, and these are believed to have tangible effects in remineralizing early tooth-enamel caries lesions. In vitro tests were conducted to compare the remineralization capacity of nHAp, with preparations containing fluorine and NovaMin [60]. All three compounds have been shown to effectively remineralise artificially produced caries, with nHAp showing significantly better results, which encourages its use in the treatment of early caries lesions [61].

Although the application of nHAp to reduce dental caries is still a relatively new approach in restorative dentistry, there are authors confirming their clinical versatility, including antibacterial activity and remineralisation [62,63,64,65,66,67]. This means that, regarding the experimental data conducted by multiple scientists, nHAp is promising to be used soon as a standard biomaterial in dentistry, including paedodontics.

### 3.3. Stem Cells

Stem cells are unspecialized cells of the body. Due to their totipotential character, they can differentiate into any cell of the organism and have the ability to self-renew. Each stage of differentiation lowers their specialization abilities [68,69]. At first, they are totipotent, and lastly, after several stages, they become unipotent stem cells that are able to differentiate into only one specific cell type.

Teeth are a very difficult material for regenerative medicine. Dental tissue recreation poses a challenge because of the function of teeth in such aspects as articulation or aesthetics. Currently, the missing tooth structure is replaced with artificial materials, since the ability of adult human dental tissues to regenerate is practically non-existent, especially in enamel, due to the absence of ameloblasts in the formed teeth. It makes the development of new techniques for regenerating lost tooth structures particularly beneficial for society [70].

One population of stem cells—hematopoietic stem cells—form all kinds of blood cells in the organism. Postnatal stem cells were already isolated from several types of human tissue, including hair follicles, skeletal muscles, the brain, and dental pulp [71,72,73]. Medical research shows that it is possible to differentiate bone marrow stem cells into ameloblasts [74]. In the case of dentin, regeneration can only occur when healthy pulp tissue is still present and when bacterial contamination has been completely removed. In such a situation, it is possible to fill the prepared cavity in order to prevent bacterial contamination and cause the formation of repair dentin.

In children with incompletely formed teeth with wide-open root tips, pulp tissue can be regenerated through open root tips [68]. Dentin regeneration is usually not possible in necrotic teeth.

According to studies [75], stem cells from human exfoliated deciduous teeth (SHED) have the ability of extensive proliferation, as well as differentiation. Exfoliated primary teeth without carious lesions may present a highly effective and easily obtainable source of stem cell tissues.

Stem cells are most widely used as the “gold standard” in the regeneration of bone tissue after tooth extractions [76]. As Figure 5 confirms, they can be successfully used for a variety of purposes.

Telles et al. [75] confirm that exfoliated deciduous teeth have similarities with the umbilical cord, as they have potential for clinical applications in regenerative medicine [77,78].

Currently, the goal of stem cell therapy is to obtain stem cells from readily available sources in the human body, such as the periodontal ligament or deciduous and permanent teeth. Periodontal regeneration is considered as an operation that is biologically and technically possible, yet it remains challenging in present-day dentistry. The periodontal ligament is a highly fibrous and vascular tissue, with one of the highest turnover rates in the whole human body [79]. Periodontitis is an inflammatory disease affecting tissues supporting the tooth, leading to their malfunction, and eventually to tooth loss [80].

Current advancements in stem cell treatments present some opportunities for tissue engineering in periodontal therapy [81]. In stem cell therapy, one of the critical requirements is a delivery of ex vivo expanded progenitor cells that can be later induced to proliferate into desired tissues. Although stem cell technology for regenerative therapies is already available, due to ethical reasons, stem cell therapies are conducted mostly on animals [82,83], and relatively little is known about stem cells in in vivo biology. The truth is that, in order to effectively use stem cells in regenerative therapy, a solid knowledge of biomaterials and growth factors is necessary [84].

The periodontal ligament consists of osteoblasts, fibroblasts, cementoblasts, epithelial, endothelial cells, and a certain amount of “progenitor cells” [79]. Pejcic et al. [79] focus on an approach to periodontal regeneration that involves the incorporation of progenitor cells in a periodontal defect. There have been confirmed cases of adipose-derived and autologous bone marrow mesenchymal stem cells that were able to regenerate periodontal ligament structures [85]. Additionally, Takahashi et. al. [86] confirmed reprogrammed fibroblasts’ utility in regenerative periodontal therapy.

### 3.4. Caries Infiltration

A method designed to treat demineralization changes without disrupting the continuity of hard dental tissue is the infiltration of caries with light-cured resin, consisting of, among others, tetraethylene glycol dimethacrylate (TEGDMA). The material’s high surface tension elevates its penetration ability by capillary forces. The main purpose of this procedure is to close the microporosity of demineralized enamel, filling the intercrystalline spaces with resin, which, in effect, stabilizes the tissue structure and isolates it from the bacteria from the oral cavity environment [87,88,89]. The enamel surface subjected to the ICON^®^ infiltration system has a microhardness comparable to that of intact enamel [90].

This technique additionally allows for the elimination of the unsightly, chalky colour characteristic of initial carious changes, and enamel development disorders due to the refractive index (refraction) of the resin, such as the enamel. The effects of the treatment are visible in the clinical example pictures in Figure 6. The treatment is carried out in two stages, and consists of cleaning, drying, and etching the treated surface with 15% hydrochloric acid (for 2 min), and then applying the resin twice. First, the resin is left for 3 min and light-cured for 40 s, then left for 1 min and polymerized as above [91,92,93].

### 3.5. Fissure Sealants

The methods that inhibit the development of caries include sealing cavities and grooves on the surface of the teeth. All lateral teeth and foramina caeca in permanent upper lateral incisors are sealed immediately after the teeth erupt, increasing the probability that they are not yet affected by caries. The dental surface protected with a sealant is easier to clean and, at the same time, inaccessible to cariogenic bacteria; it is particularly advisable to protect fissures with a high risk of caries (high degree of plaque retention). Among the materials intended for coating, we can distinguish a resin-based group, e.g., Helioseal F, Fissurit, F Visioseal, and also flow composites: chemo- or light-curing, transparent or coloured, with or without fluorine and biocompatible glass ionomer materials, e.g., Fuji Triage Pink (for erupting teeth), chemically and light-cured Fuji Triage White (for erupted teeth) with a high content of fluoride. Recent studies from 2019 found that the use of nanohydroxyapatite before sealing demineralized fissures improves the adhesion of fissure-sealing materials; however, this did not have a significant effect on the marginal micro-leakage. It was found that the simultaneous use of SHMP (sodium hexametaphospate) and nHAp (0.15%) and a crevice seal can be considered a minimal intervention in dentistry for the protection of demineralized fissures and pits [94]. Crevice lacquer is a protective layer that prevents the penetration of cariogenic bacteria and facilitates the effective cleaning of cavities [95,96]. Deep carious lesions in a tooth are diagnosed when they reach three-quarters or more of the dentin depth visible on a radiological image [97]. The goals of restorative treatment include the repair or reduction of damage caused by caries, protecting and preserving the structure of the teeth, as well as maintaining the vitality of the pulp.

### 3.6. Indirect Pulp Capping

According to the AAPD, the most important goal when treating a tooth with caries is to maintain the vitality of the pulp in deciduous teeth, especially in immature permanent teeth, for further root development (apexogenesis) [98,99]. For better understanding, the method is presented in Figure 7. Depending on the treatment method, the priority is an individual approach to the patient by determining the risk of caries, understanding the caries process, and supervision to monitor its progress. In the case of deciduous teeth, the time they will remain in the oral cavity before physiological replacement should be considered. The treatment of deep caries, both in deciduous and immature permanent teeth, according to the AAPD, involves indirect pulp capping, (IPC), including single-stage or multi-stage caries removal, provided that the tooth is asymptomatic or responds correctly to vitality tests, and that, in the case of reversible pulpitis, pain does not persist after thermal or mechanical provocation, and no root resorption is found radiographically [97].

The one-step treatment of caries involves the almost-complete removal of the infected dentin, that is, the complete removal of carious masses from the walls except for the cavity bottom, where a thin layer of demineralized dentin is left for remineralization. Leaving a thin layer of demineralized dentin protects the pulp against mechanical damage during cavity preparation. One-step IPC is performed when there is no plan to reopen the cavity and remove any remaining infected dentin [101]. The bottom of the cavity is provided with a highly alkaline antibacterial material, and then with a final tight filling [99]. The literature shows that the one-step IPC method in deciduous teeth shows better treatment results, confirmed radiographically, compared to pulpotomy, i.e., the complete removal of pulp [102], with symptoms such as pain or swelling and the absence of any abnormalities on the radiography of the treated tooth [103]. Pinto et al. [104], as well as Bressani et al. [105], assessed the clinical effects of one-step IPC in deciduous teeth after the application of calcium hydroxide, examining incompletely removed caries. The data show that after 4–7 months, the remaining dentin was hard and dry, and there was also a significant reduction in the level of aerobic and anaerobic bacteria [104]. In contrast, Pinto et al. found that, after 3 months, the dentin became harder and darker [105].

### 3.7. Staged Caries Removal

Another method of treating deep caries is their gradual removal. This method is considered a safe procedure used in deciduous and immature permanent teeth and involves two visits: during the first one, the carious masses are completely removed from the cavity walls, while only soft masses are removed from the cavity bottom [101]. Then, a highly alkaline antibacterial material is applied to the bottom, which is sealed tightly with glass ionomer cement. During the second visit, carious debris is removed, and the cavity is finally closed. According to the AAPD guidelines, a suitable time for tertiary dentin formation is an appointment interval of 3–6 months [99]. The criteria for the effectiveness of the above-mentioned method are the absence of symptoms such as sensitivity, pain, or swelling. Moreover, the reaction of pulp to thermal tests of the treated tooth is correct, and radiographically, there are no periapical changes, and a further development of the root of immature permanent teeth is visible [101,106]. Both methods are recommended for the treatment of deep caries compared to complete caries removal, which is more aggressive and can result in pulp damage. However, there is also evidence that the partial (single step) method is preferable to the stepwise (two-step) method. A review of the Cochrane databases showed that both the partial and gradual removal of carious masses from the cavity reduced the risk of pulp damage in asymptomatic deciduous and permanent teeth [107]. However, there is also evidence that the partial (one-step) method is superior to the stepwise (two-step) method, as reopening the cavity may cause accidental pulp damage. In both methods, success depends on the tightness of the filling, as a well-sealed cavity inhibits the flow of nutrients to the bacteria remaining in the infected dentin [108]. The materials that are used in both methods should be characterized by, e.g., biological compatibility, adequate resistance to occlusive forces, anti-caries properties, and properties limiting biofilm deposition.

### 3.8. MTA, Biodentine, and Deep Caries

Modern dentistry uses materials with high tissue biocompatibility and very good adhesion to dentin, hydrophilic sealing strips, and materials showing X-ray contrast thanks to metal particles. These materials include bioactive dentin substitutes such as Mineral Trioxide Aggregate (MTA) and Biodentine cement by Septodont. The MTA conglomerate has an antibacterial effect; it is also active against anaerobes and Enterococcus faecalis and is antifungal [109]. Research has shown that MTA cement has undergone biocompatibility tests which demonstrated, after application, marginal tightness through adaptation to the hard tissues of the tooth, durability, and permanent tightness [110,111]. These materials are used for supplying root perforation and the bottom of the pulp chamber exposure, as well as for internal resorptions. They also support the apexification processes of the root apex in immature permanent teeth. The stimulation results in the production of a hard, very dense structure and a much greater strength than the compared stimulation with calcium hydroxide preparations. As it was found in the study, teeth with MTA preparations were more resistant to fractures than teeth with calcium hydroxide preparations [112,113]. MTA preparation is proven to induce the processes of cementogenesis and osteogenesis by inducing neutrophil chemotactic factor substances, including interleukin 1 beta (IL-1beta) and Macrophage Inflammatory Protein 2 (MIP-2) [114]. The research by Reyes-Carmon et al. on animal tissues demonstrated its biomineralizing and regenerative properties [109,111,112]. MTA is biologically compatible with host tissues. This phenomenon is related to the deposition of hydroxyapatite crystals on the surface of the material and the tissue-regeneration material. MTA is also classified as a cement-forming material due to the induction of osteoblasts on its surface and the secretion of genes encoding the protein [113]. In the case of deep caries, a highly alkaline antibacterial material is applied to the remaining thin layer of demineralized dentin. For this purpose, calcium hydroxide, zinc oxide with eugenol, mineral trioxide aggregate (MTA), or Biodentine can be used. In order to remineralize dentin and protect the pulp from the toxic components of resin-based materials, calcium hydroxide materials have been widely used. Currently, MTA and Biodentine are the materials of choice, which are a very good alternative to the calcium hydroxide-based formulations used so far. MTA and Biodentine are biocompatible and bioactive materials. The MTA material is mainly composed of tricalcium and dicalcium silicates. It is a strongly alkaline preparation; its pH is 10.2 immediately after binding and increases to approximately 12.5 after a few hours [111]. Thanks to this, MTA effectively inhibits the growth of microorganisms and inflammatory processes in the living pulp and stimulates mineralization processes [111]. However, the disadvantages of this material are the long setting time of 2 h 45 min and poor mechanical properties [115]. In contrast, Biodentine, whose main component is tricalcium silicate and dicalcium silicate, was developed to form repair dentin for replacing dentin in both the crown and root regions. Many in vitro and in vivo studies have shown that the interactions of Biodentine with hard and soft tissues ensure tight adhesion to the dentin, protecting the tooth pulp by preventing bacterial infiltration. These studies also showed that, by interactions with hard tissues, Biodentine provides micromechanical retention by infiltrating tubules [116]. On the other hand, it induces the synthesis of tertiary dentin, which provides further pulp protection. These two combined effects may be responsible, in part, for the lack of post-operative pain and hypersensitivity [115].

Compared to MTA, Biodentine is easier to apply, has strong mechanical properties and a shorter setting time, and can fill the entire cavity temporarily for 6 months [115]. In contrast to calcium hydroxide, Biodentine has stronger mechanical properties due to its composition and lower porosity and is less soluble, which means that it has a better sealing ability [116], which can be seen in Figure 8. 

In the one-step method, in the case of using calcium hydroxide or MTA before placing the final (composite) fill, these materials should be secured with glass ionomer cement, while materials based on resins can be directly applied to Biodentine [115]. Biodentine was introduced to the market in 2011. The preparation consists of a powder and a liquid. The powder consists of tricalcium silicate responsible for the setting process, calcium carbonate for filling and improving mechanical properties, zirconium dioxide responsible for contrast on the X-ray, dicalcium silicate, calcium, and iron oxides. The fluid is an aqueous solution of calcium chloride and a copolymer that reduces the viscosity of the cement. After combining the components, the cement has a high content of calcium hydroxide and thus has alkaline properties. The material is also biocompatible with dental tissues, shows high hardness and marginal tightness, and does not dissolve in saliva. However, a rather considerable disadvantage is its sensitivity to moisture and poor X-ray contrast. After application to the tooth, the preparation creates a zone of mineral infiltration in contact with dentin and enamel. This results in collagen degeneration, facilitates the penetration of Ca^2+^, OH^−^, CO_3_^2−^ ions, and thus increases the mineralization of dental tissues. Indications for the use of Biodentine are numerous: supplying the perforation of the root wall and the bottom of the pulp chamber, internal and external resorption, and the stimulation of apexification of the root apex of an immature permanent tooth with direct and indirect pulp capping [118,119].

### 3.9. Glass Ionomer Cement

Materials that have found widespread use in paediatric dentistry as temporary materials or for final tooth reconstruction are glass ionomer cements, resin-modified glass ionomer cements, and composites that are used for the final restoration of the tooth. Today, nanotechnology-oriented research is emerging more and more, especially in the field of dentistry, which focuses on the potential applications and benefits of nanomaterials over the conventionally used materials. Nanotechnology is the art and science of materials science on a scale of less than 100 nm, and it is getting more and more attention by improving the mechanical and physical properties of materials [120].

Conventional glass ionomer cement, used since the late 1970s, contains aluminium-fluorosilicate glass (FAS), which has bioactive properties due to the presence of silicates and fluorides. It is a translucent, water-based cement. Bioactivity induces cell growth, proliferation, and tissue formation by the biomaterial. Moreover, bioactivity also influences the antibacterial activity of the material and thus prevents or treats infections in the tissues [121].

The advantages of traditional glass ionomer cements include the ability to chemically bind to dental tissues by chelation of the carboxyl group of polymer chains of acid and calcium ions in enamel and dentin apatites. In addition, these materials have acceptable translucency and colour, and exhibit an anticaries effect due to the release of fluoride. The most important advantages of nano-glass ionomer cements are increased wear resistance and very good aesthetics, as well as the anticariogenic effect of fluoride, leading to the inhibition of bacterial growth and metabolism in the oral cavity environment [122]. The clinical indications are the temporary and final reconstruction of deciduous teeth, small class I restorations, reconstruction with the open and closed sandwich technique, and class III and V restorations [123].

A disadvantage of traditional glass ionomer cements is their limited scope of application. Due to their poor mechanical properties and low load-capacity (cracking, breaking), they are most often used in molars as a temporary filling [123]. Another disadvantage is that they are sensitive to moisture immediately after application [121]. To improve the mechanical properties of the material, some modifications were introduced, such as combining conventional glass ionomer cements with resins.

Moreover, since the first use of glass ionomer cement, research has been focused on improving its properties. As recent research suggests that the inclusion of nano-sized particles may improve the mechanical properties of restorative dental material such as composites, [124] a similar attempt was made to improve the physical and mechanical properties of glass ionomer cements with the use of nanotechnology [121]. For this purpose, various types of nano-sized powders, such as hydroxyapatite and fluorhydroxyapatite, that are similar to mineralized dental tissues and initially used in the inhibition of caries, were added to glass ionomer cements [125]. An example may be nanohydroxyapatites (nHAp), which promote enamel remineralization. [126] Moreover, the addition of nHAp to powder components of a conventional glass ionomer cement increases the crushing, tensile, and bending strength after storage in distilled water for 7 days [127]. Furthermore, glass ionomers containing nanoapatites are expected to bond better to the dental surface due to the possibility of forming strong ionic bonds between the apatite crystals/particles in the cement and the calcium ions in the dental structure.

The amount of fluoride released by nano-glass ionomer cements is a questionable issue. Several studies show that the release of fluorine from nano-glass ionomer cements and conventional resin-modified glass ionomer cements is comparable, but still much lower than that of conventional glass ionomer cements [128,129].

Recently, there were attempts to add bioactive glass in order to enhance the bioactivity of the glass-ionomer cements [130]. Combinations of bioactive glass with 10mol% Al^3+^ added in ≤20 wt% to a GIC have good biocompatibility and show the highest strength of the material.

Despite the great effectiveness of the infiltration method, there are some studies that emphasize the poor biocompatibility of the material. It has been noticed that TEGDMA can disturb the pulp’s homeostasis and regeneration potential [131,132].

### 3.10. Composite Material

Another material used for filling both deciduous and immature permanent teeth are composites (composite materials), which consist of a matrix (polymer resin), an inorganic filler, and a dye. Depending on the size of filler particles, they are divided into macrophilic, midiphilic, miniphilic, microfilm, and nanophilic particles. Recent studies describe the development of a nano-amorphous composite resin filled with nano-amorphous calcium phosphate [133]. Nanoparticles can improve the remineralizing properties of composites as well as maintain the same level of calcium and phosphorus release [134]. Moreover, due to the ability to quickly neutralize bacterial acids through the release of calcium and phosphorus, this material can inhibit the initiation of secondary caries [135]. In order to prevent the initiation and progression of secondary caries, the composites additionally use a nanocomposite coating consisting of lactose-modified chitosan with silver nanoparticles (nAg) [136]. Ionescu et al. found in their studies that the finely dispersed nanoparticles in the nanocomposite coating were able to significantly reduce the formation of biofilm on the filling surface by 80% after 48 h of use, compared to the control without the coating. Images obtained with the use of Confocal Laser Scanning Microscopy (CLSM) showed that the coating had no bactericidal activity and showed the ability to change biofilm morphology, preventing the development of mature biofilm species (65). Endodontic treatment is a consequence of pulp diseases or tooth injuries and is aimed at maintaining the tooth in the mouth as long as possible. However, the treatment process may be subject to iatrogenic complications, such as the perforation of the root wall or the bottom of the pulp chamber. Perforation is defined by the American Association of Endodontics (AAE) as mechanical or pathological disruption in the continuity between the tooth canal system and the outer surface of the root. Iatrogenic errors are the second most common endodontic complications (9.6%) and arise, for example, as a result of the use of an incorrectly sized endodontic instrument for canal dilation or the mechanical preparation of curved canals. The prognosis for maintaining a tooth with a perforation in the oral cavity depends on the time it takes to supply the perforation, the extent and location of the perforation, and the material it is supplied with. In the last century, zinc oxide with eugenol, calcium hydroxide, amalgam, and tricalcium phosphate were used to supply perforations. These materials did not meet the expectations, as they bonded poorly in a humid environment, did not show an adequate tightness of bonding with the dental tissues, deformed after concentration, and often were difficult to apply at the perforation site [110].

### 3.11. Atraumatic Restorative Techniques

Dental anxiety is a common worldwide phobia being present in around 12–20% of the population around the globe [137]. This fear is generally even more common in paediatric patients than it is in adults. Not only is the unpleasant treatment procedure itself, with rotating burs, noises, smells, and vibrations difficult for a patient, but so is an anaesthetic injection, which acts as a pain relief. Ironically, despite its immediate effect, the injection itself is such an unpleasant experience that it has become yet another significant anxiety trigger during dental treatment [138].

For this reason, modern dentistry continues to develop less and less traumatic, yet effective, dental restoration procedures. Chemo-mechanical caries removal [139] and preparation using air abrasion and lasers [140] are some of the most noteworthy techniques currently in development in modern atraumatic dentistry.

## 4. Effect of Dental Biomaterials Application on the Progress of Oral Inflammatory Reactions

Periodontal disease is a disorder occurring by the effect of an imbalance between the immune system of the oral cavity and microorganisms. When untreated, it eventually leads to local disorders like tooth structure support and/or alveolar bone loss. Although local effects of periodontitis are directly dangerous for the patient, there are also articles confirming its impact on general health [141,142], causing medical conditions such as chronic kidney disease or coronary heart disease [143].

The aetiology of periodontitis includes the action of macrophages and neutrophils, as well as proinflammatory mediators such as interleukins, prostaglandins, or C-reactive proteins that are released to the bloodstream after the exposure to periodontal bacteria.

Galectins are beta-galactoside-binding proteins present in active stages of inflammation [144]. Galectin-3 is one example of these proteins that is systemically released on active inflammation sites, and its levels appear to be elevated in conditions such as coronary heart disease or heart failure [145]. Isola et. al. [146] confirmed in their study that patients with periodontitis and coronary heart disease had higher levels of Galectin-3 when compared to patients with coronary heart disease alone or healthy controls.

The fact is that one of the major factors influencing the initiation of gingivitis and periodontitis is hard dental-tissue caries [147]; the successful application of biomaterials and the achieved hard dental-tissue regeneration proves to be one of the many methods of periodontitis formation prevention.

## 5. Conclusions

The use of nanotechnology is constantly evolving, and although nanomaterials offer excellent aesthetics and polishing, their mechanical properties are inferior to, for example, microfilled composites [133]. Due to the molecular scale, nanohydroxyapatite exhibits unique properties, especially in terms of the accuracy of tissue reconstruction. It should be used in paediatric dentistry as a component of oral hygiene products with a remineralizing effect, and further research—aimed at broadening the knowledge and furthering the understanding of clinical interactions—should focus primarily on its use in endodontic treatment, in direct pulp capping, perforation supply, and apical barrier formation after the treatment of immature permanent teeth, and its use as components of glass-ionomer cements and composite materials. In addition to the use of nanohydroxyapatite, there are many methods of reconstructing hard dental tissues, such as the use of infiltration, sealing, indirect filling, stem cells, or glass ionomer cement. In each case, the procedure is aimed at replacing natural tissues with artificial ones or remineralizing them. The development of such technologies and methods allows for an increasing range of dental-tissue regeneration services.

## Figures and Tables

**Figure 1 nanomaterials-11-03374-f001:**
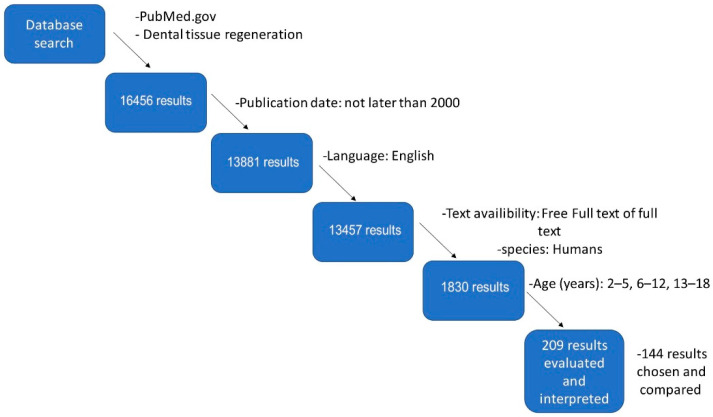
Methods of database search in PubMed.

**Figure 2 nanomaterials-11-03374-f002:**
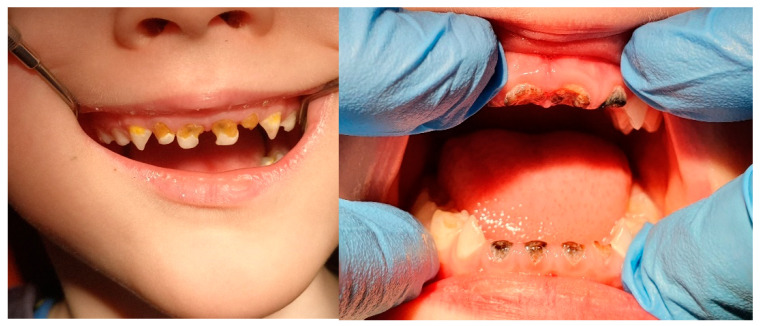
Clinical examples of early childhood caries (ECC). Source: Patient of the Wroclaw Medical University, Department of Pediatric Dentistry and Preclinical Dentistry, ul. Krakowska 26, Wroclaw.

**Figure 3 nanomaterials-11-03374-f003:**
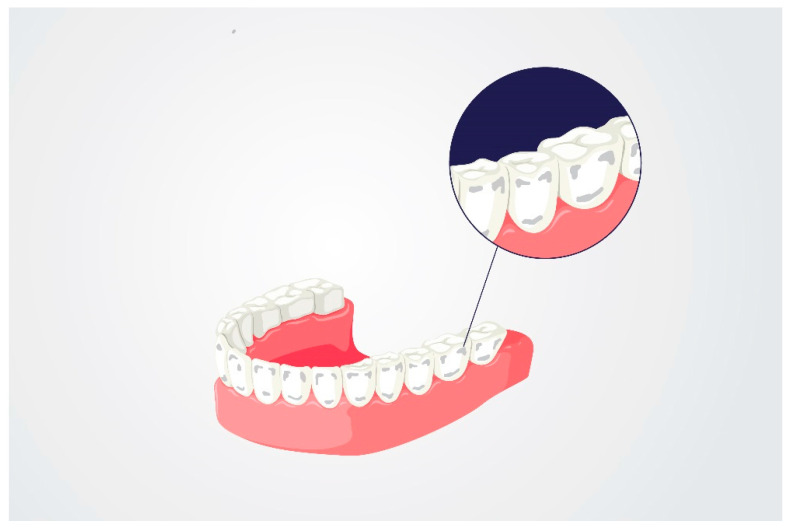
Representation of the most common white spot lesion (WSL) locations. Adapted from Ref. [30].

**Figure 4 nanomaterials-11-03374-f004:**
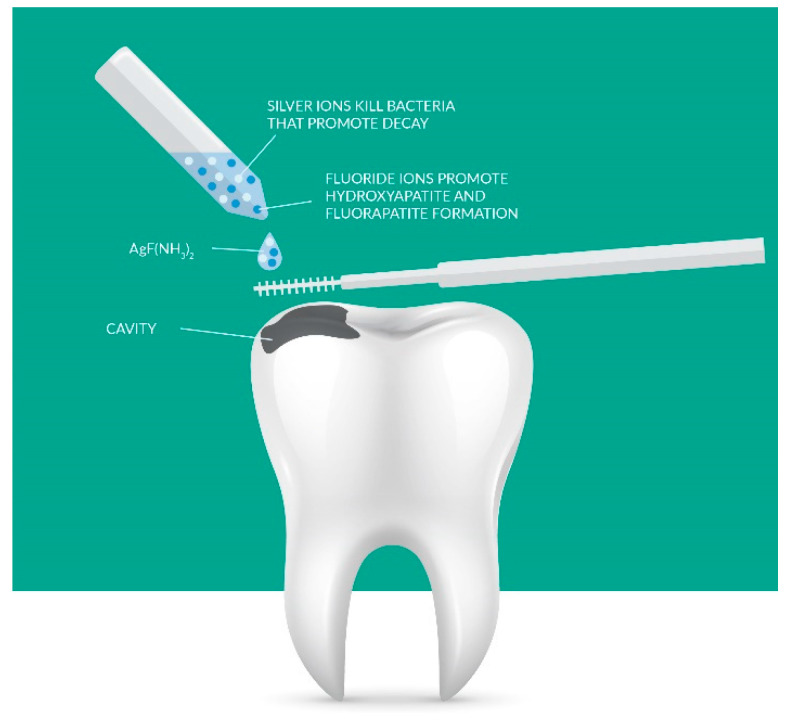
Applying materials containing silver and fluoride ions promotes the antimicrobial effect of tooth remineralization. Adapted from Ref. [51].

**Figure 5 nanomaterials-11-03374-f005:**
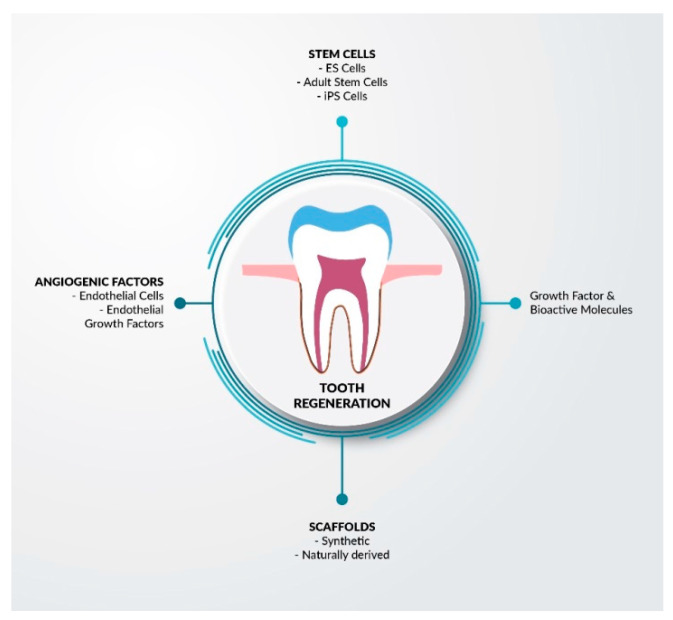
Possible applications of stem cells in the process of dental tissue regeneration. Adapted from Ref. [68].

**Figure 6 nanomaterials-11-03374-f006:**
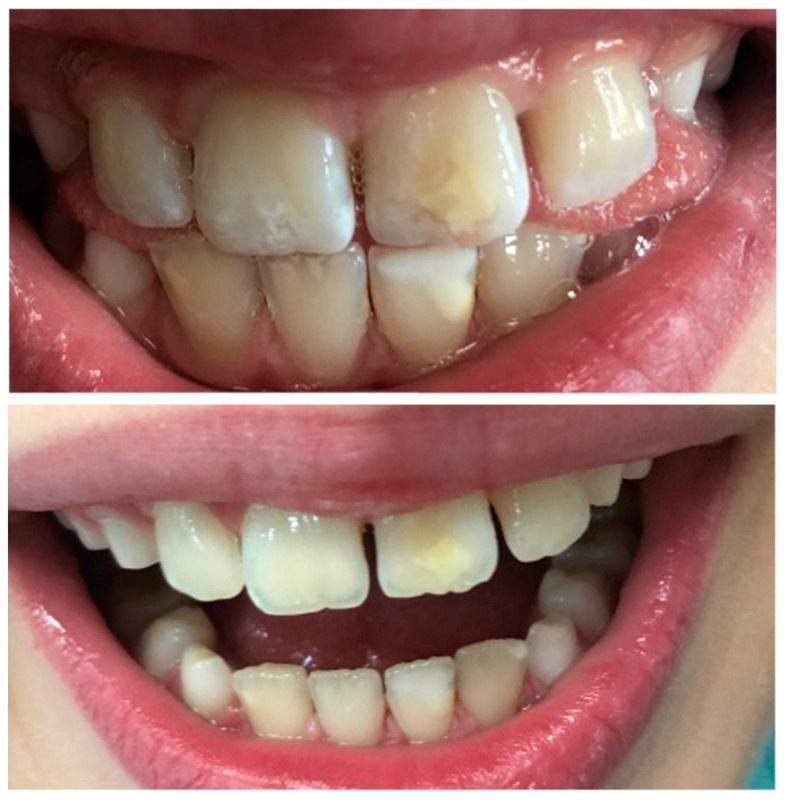
Comparison of caries infiltration with ICON^®^, before (**top**) and after the procedure (**bottom**). Source: Patient of the Wroclaw Medical University, Department of Pediatric Dentistry and Preclinical Dentistry, ul. Krakowska 26, Wroclaw.

**Figure 7 nanomaterials-11-03374-f007:**
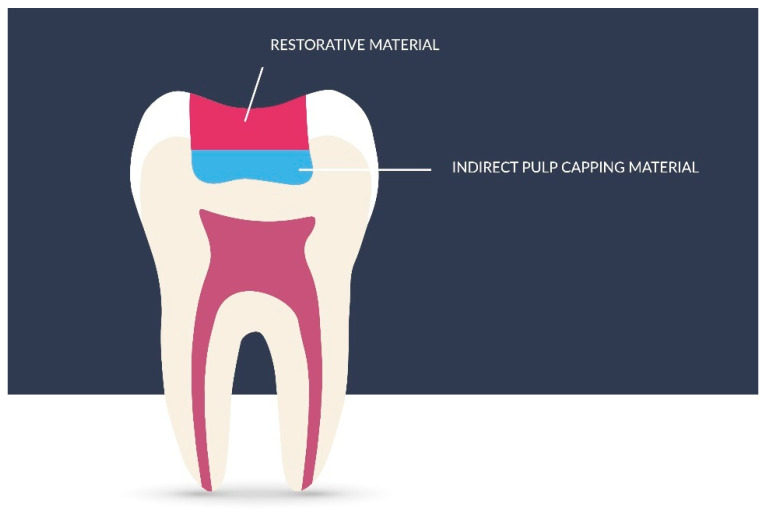
Graphical representation of indirect pulp capping method. Adapted from Ref. [100].

**Figure 8 nanomaterials-11-03374-f008:**
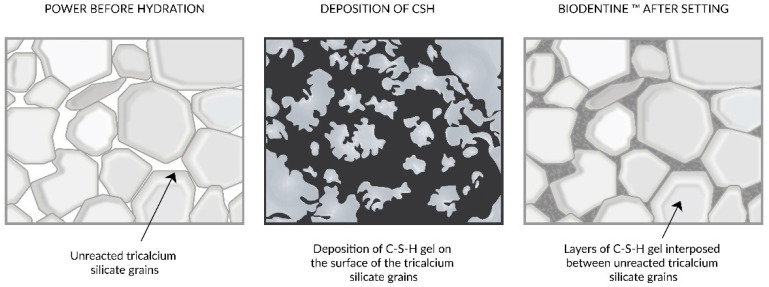
Sealing ability of setting Biodentine. Microscopically visible spaces between unreacted tricalcium silicate grains (**left**), deposition of C-S-H gel on silicate grains (**middle**), C-S-H gel filling spaces between unreacted tricalcium silicate grains (**right**). Adapted from Ref. [117].

## Data Availability

Not applicable.

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
