# Peer review of "Application of Selected Biomaterials and Stem Cells in the Regeneration of Hard Dental Tissue in Paediatric Dentistry—Based on the Current Literature"

_nanomaterials, 2021, doi:10.3390/nano11123374_

Round 1
Reviewer 1 Report
After carefully reading the manuscript, I must admit to having difficulties understanding what exactly this paper is about. Many different concepts are randomly mixed together in a text that does not appear to have a logical flow and coherent structure. On many occasions, the manuscript has the character of a dental textbook (describing in detail how to perform a certain dental procedure), and dental materials textbook (discussing characteristics of individual dental materials), then it unexpectedly introduces and documents the clinical experience of the authors (as in a case report), and on top of that, the concept of “nano” is only vaguely tackled, as if only to mention the topic but without integrating it logically in the text. Additionally, some historical facts about materials are randomly incorporated in the text, together with data taken from instructions for use of commercial dental products. All of this produces a long text that is difficult to follow and extract useful data.
For the most part, the manuscript looks like an essay, describing the flow of thoughts of the authors. Being mostly clinically oriented, that essay has no connection to the topic of nanomaterials. Procedures of caries removal, application of stainless-steel crowns, classifications of tooth caries, etc. belong to the field of clinical dentistry and not to nanomaterials. Only occasionally, the idea of nanomaterials is mentioned and it seems like it has been forcefully inserted in the text, interrupting its logical flow.
For example, section 3.2 begins with the discussion on fluoride for caries prevention, then suddenly changes the direction towards biological effects of restorative materials, then introduces “biomimetic hydroxyapatite” (please note that the chemical formula for hydroxyapatite is incorrect!) as a therapeutic agent, only to discuss enamel structure several sentences later, then going to quantum-physical effects of nano-sized materials, which is followed by historical remarks about several commercial products. This type of writing generally occurs throughout the whole manuscript; the reader is left puzzled about the point of the essay. Most of the presented information is not state-of-the-art but rather common knowledge of a practicing dentist. A revised version of this manuscript (with improved logical flow) would be more appropriate for a student lecture transcript than for a scientific journal. The scientific contribution is minor at best, as it is very difficult for the reader to navigate through the text to find some novel useful information.
The manuscript does not seem to follow what is presented in the Materials and methods section. The search strategy for the literature review seems to be “invented” after the manuscript was already written, in order to satisfy the form. It is unclear what the search strategy was, as the description in the text contradicts what is shown in Figure 1. A simple example that highlights the inconsistency: if “dental tissue regeneration” were the only search keywords (which is also an inappropriate search strategy), how did the authors end up discussing stainless steel crowns, while missing some more important aspects of regenerative dentistry?
Also, some minor points:
“Biodentine” is used interchangeably with “Betadine”
Please use the term “bioactive” materials more conservatively, in accordance to the latest recommendations
Author Response
Dear Reviewer, We would like to express our sincerest gratitude to the Reviewers for their enormous efforts in criticizing the manuscript. All remarks have been included in the revised version of the manuscript.
Question 1
After carefully reading the manuscript, I must admit to having difficulties understanding what exactly this paper is about. Many different concepts are randomly mixed together in a text that does not appear to have a logical flow and coherent structure. On many occasions, the manuscript has the character of a dental textbook (describing in detail how to perform a certain dental procedure), and dental materials textbook (discussing characteristics of individual dental materials), then it unexpectedly introduces and documents the clinical experience of the authors (as in a case report), and on top of that, the concept of “nano” is only vaguely tackled, as if only to mention the topic but without integrating it logically in the text. Additionally, some historical facts about materials are randomly incorporated in the text, together with data taken from instructions for use of commercial dental products. All of this produces a long text that is difficult to follow and extract useful data.
Answer: We would like to thank you for the comment. The article is meant to review current approach to the use of biomaterials in hard dental tissue restoration. The idea is to introduce the reader to the possible techniques of dental tissue restoration, with addition of nanomaterials as one of the possible restoration materials. Additionally, the manuscript is meant to confirm the possible techniques’ use with clinical examples. The historical facts have been corrected.
Question 2
For the most part, the manuscript looks like an essay, describing the flow of thoughts of the authors. Being mostly clinically oriented, that essay has no connection to the topic of nanomaterials. Procedures of caries removal, application of stainless-steel crowns, classifications of tooth caries, etc. belong to the field of clinical dentistry and not to nanomaterials. Only occasionally, the idea of nanomaterials is mentioned and it seems like it has been forcefully inserted in the text, interrupting its logical flow.
Answer: We would like to thank you for the comment. The manuscript is a review of current approach to the use of biomaterials, with nanomaterials being one of the clinical possibilities in hard dental tissue restoration. The stainless-steel crowns subchapter has been removed.
Question 3
For example, section 3.2 begins with the discussion on fluoride for caries prevention, then suddenly changes the direction towards biological effects of restorative materials, then introduces “biomimetic hydroxyapatite” (please note that the chemical formula for hydroxyapatite is incorrect!) as a therapeutic agent, only to discuss enamel structure several sentences later, then going to quantum-physical effects of nano-sized materials, which is followed by historical remarks about several commercial products. This type of writing generally occurs throughout the whole manuscript; the reader is left puzzled about the point of the essay. Most of the presented information is not state-of-the-art but rather common knowledge of a practicing dentist. A revised version of this manuscript (with improved logical flow) would be more appropriate for a student lecture transcript than for a scientific journal. The scientific contribution is minor at best, as it is very difficult for the reader to navigate through the text to find some novel useful information.
Answer: We would like to thank you for the comment. The subchapter has been modified according to the reviewers suggestions. The hydroxyapatite formula has been corrected.
Question 4 The manuscript does not seem to follow what is presented in the Materials and methods section. The search strategy for the literature review seems to be “invented” after the manuscript was already written, in order to satisfy the form. It is unclear what the search strategy was, as the description in the text contradicts what is shown in Figure 1. A simple example that highlights the inconsistency: if “dental tissue regeneration” were the only search keywords (which is also an inappropriate search strategy), how did the authors end up discussing stainless steel crowns, while missing some more important aspects of regenerative dentistry?
Answer: We would like to thank you for the comment. The Stainless steel crown subchapter has been removed. The search strategy has been updated, with clear number of evaluated and compared results.
Question 5 “Biodentine” is used interchangeably with “Betadine”
Answer: We would like to thank you for the comment. The mistake has been corrected.
Reviewer 2 Report
Dear Authors.
The title is not suitable for this current paper. Could you please revise it and create a suitable one for the whole manuscript.
Keywords: are quite poor. Try to revise them.
Abstract: is weak. revise it.
Introduction: Below is a few suggestions for improvement of this paper.
a) Lie-50: add the source of this statement.
b) In the introduction try to add some references at the end of the statements. I am suggesting a few references suitable. Will help authors;
"The approach of regenerative dentistry..... of healing and the different stages of tooth development" end of this line cite this paper "Zafar, M. S., Khurshid, Z., & Almas, K. (2015). Oral tissue engineering progress and challenges. Tissue Engineering and Regenerative Medicine, 12(6), 387-397."
In heading 2: The whole method part need to be cited what is the protocol they use. A very recent paper published and reported the way of extracting the dental literature. I am recommending authors read this paper and improve search strategy and MeSH terms. After reading this paper they can use it as a source and cite it. "Khurshid, Z., Tariq, R., Asiri, F. Y., Abid, K., & Zafar, M. S. (2021). Literature search strategies in dental education. Journal of Taibah University Medical Sciences.
Figure-1: resolution is not good.
Line 80-100: need serious attention from the expert to read and revise. What are the sources of this information? I have noticed authors not providing the sources or references in many places.
Heading 3.1: long statements with strong information but no sources or citations.
In heading 3.2: Please read this paper and include information in the lines." Chen, Lijie, et al. "Hydroxyapatite in Oral Care Products—A Review." Materials 14.17 (2021): 4865.".
Figure-4: authors have to declare the source or mention the papers used for designing this work.
In heading 3.1: the authors mentioned many commercially available dental care products but they do not use trademark or city or country. Please carefully mention all products names.
Figure-5 and 6 need proper sources.
Heading 3.9:
Again authors forget to cite the references or they are not aware of scientific writing. Add information from this paper. https://doi.org/10.3390/ma14216260
Hope all the above suggestion can help authors for the improvement and resubmission.
Author Response
Dear Reviewer,
we would like to express our sincerest gratitude to the Reviewers for their enormous efforts in criticizing the manuscript. All remarks have been included in the revised version of the manuscript.
The title is not suitable for this current paper. Could you please revise it and create a suitable one for the whole manuscript.
Answer: We would like to thank you for the comment, the title has been modified according to the Reviewer’s suggestions.
- Keywords: are quite poor. Try to revise them.
Answer: We would like to thank you for the comment. The keywords have been corrected.
- Abstract: is weak. revise it.
Answer: We would like to thank you for the comment. The Abstract has been connected
- Lie-50: add the source of this statement.
Answer: We would like to thank you for the comment. The source has been added
- In the introduction try to add some references at the end of the statements. I am suggesting a few references suitable. Will help authors;
Answer: We would like to thank you for the comment. The references have been added.
- "The approach of regenerative dentistry..... of healing and the different stages of tooth development" end of this line cite this paper "Zafar, M. S., Khurshid, Z., & Almas, K. (2015). Oral tissue engineering progress and challenges. Tissue Engineering and Regenerative Medicine, 12(6), 387-397."
Answer: We would like to thank you for the comment. The reference has been added.
- In heading 2: The whole method part need to be cited what is the protocol they use. A very recent paper published and reported the way of extracting the dental literature. I am recommending authors read this paper and improve search strategy and MeSH terms. After reading this paper they can use it as a source and cite it. "Khurshid, Z., Tariq, R., Asiri, F. Y., Abid, K., & Zafar, M. S. (2021). Literature search strategies in dental education. Journal of Taibah University Medical Sciences.
Answer: We would like to thank you for the comment. The protocol of database search has been updated according to the Reviewer’s suggestions.
- Figure-1: resolution is not good.
Answer: We would like to thank you for the comment, Figure 1 has been modified.
- Line 80-100: need serious attention from the expert to read and revise. What are the sources of this information? I have noticed authors not providing the sources or references in many places.
Answer: We would like to thank you for the comment. Mentioned chapter of the manuscript has been corrected according to the reviewer’s suggestions.
- Heading 3.1: long statements with strong information but no sources or citations.
Answer: We would like to thank you for the comment. Mentioned chapter of the manuscript has been corrected according to the reviewer’s suggestions.
- In heading 3.2: Please read this paper and include information in the lines." Chen, Lijie, et al. "Hydroxyapatite in Oral Care Products—A Review." Materials 14.17 (2021): 4865.".
Answer: We would like to thank you for the comment. Mentioned chapter of the manuscript has been corrected according to the reviewer’s suggestions. The paper has been added
- Figure-4: authors have to declare the source or mention the papers used for designing this work.
Answer: We would like to thank you for the comment. The source has been added to the figure.
- In heading 3.1: the authors mentioned many commercially available dental care products but they do not use trademark or city or country. Please carefully mention all products names.
Answer: We would like to thank you for the comment. The manuscript has been modified according to Reviewer’s suggestions.
- Figure-5 and 6 need proper sources.
Answer: We would like to thank you for the comment. The sources have been added.
- Again authors forget to cite the references or they are not aware of scientific writing. Add information from this paper. https://doi.org/10.3390/ma14216260
Answer: We would like to thank you for the comment. The manuscript has been modified according to Reviewer’s suggestions.

Reviewer 3 Report
This is a fairly interesting narrative review of the literature on biomaterials used in pediatric dentistry.
Some serious criticisms are present
-It should be noted that this is a narrative review of the literature in the title
-The section on materials and methods is definitely problematic; the PICO strategy, the elements of inclusion and exclusion must be indicated in detail.
Furthermore, all search strings with MESH terms must be expressly indicated in the text or with a table.
No mention of the quality assessment was performed
-In the treatment of carious lesions (3.) it is necessary to indicate, even only briefly, the new innovative treatment strategies. In this regard, I recommend including the following scientific work in the reference section:
Cianetti S, Abraha I, Pagano S, Lupatelli E, Lombardo G. Sonic and ultrasonic oscillating devices for the management of pain and dental fear in children or adolescents that require caries removal: a systematic review. BMJ Open. 2018 Apr 28; 8 (4): e020840. doi: 10.1136 / bmjopen-2017-020840.
-The section relating to glass ionomer cements should be enriched with attempts in the literature to add bioactive substances that could improve the clinical performance of the cement
Author Response
Dear Reviewer,
we would like to express our sincerest gratitude to the Reviewers for their enormous efforts in criticizing the manuscript. All remarks have been included in the revised version of the manuscript.
- It should be noted that this is a narrative review of the literature in the title
Answer: We would like to thank you for the comment. The narrative aspect of the scientific work has been included in the text.
- The section on materials and methods is definitely problematic; the PICO strategy, the elements of inclusion and exclusion must be indicated in detail.
Furthermore, all search strings with MESH terms must be expressly indicated in the text or with a table.
Answer: We would like to thank you for the comment. The materials and methods section has been modified.
- -In the treatment of carious lesions (3.) it is necessary to indicate, even only briefly, the new innovative treatment strategies. In this regard, I recommend including the following scientific work in the reference section:Cianetti S, Abraha I, Pagano S, Lupatelli E, Lombardo G. Sonic and ultrasonic oscillating devices for the management of pain and dental fear in children or adolescents that require caries removal: a systematic review. BMJ Open. 2018 Apr 28; 8 (4): e020840. doi: 10.1136 / bmjopen-2017-020840.
Answer: We would like to thank you for the comment. Scientific work has been added to the manuscript.
- The section relating to glass ionomer cements should be enriched with attempts in the literature to add bioactive substances that could improve the clinical performance of the cement
Answer: We would like to thank you for the comment. Additional information have been added to the manuscript.

Round 2
Reviewer 1 Report
I would like to thank the authors for providing the revised version of the manuscript. Although I still think the manuscript is slightly below the standards required for publication in a highly-ranked journal such as Nanomaterials, I have no further remarks and suggest that this submission proceed to Editor's decision.
Author Response
Dear Reviewer, we would like to thank you for the comment and express our sincerest gratitude to the Reviewers for their enormous efforts in criticizing the manuscript.
Reviewer 2 Report
Dear Author
Paper is well revised. Just check any grammatical mistakes.
Author Response

(The authors gave the same response as above.)

Reviewer 3 Report
all comments were added
Author Response
Dear Reviewer, we would like to thank you for the comment and express our sincerest gratitude to the Reviewers for their enormous efforts in criticizing the manuscript.
This manuscript is a resubmission of an earlier submission. The following is a list of the peer review reports and author responses from that submission.
Round 1
Reviewer 1 Report
Review
Comparison of the application of selected biomaterials promoting the regeneration of hard dental tissue in pediatric dentistry
The manuscript “Comparison of the application of selected biomaterials promoting the regeneration of hard dental tissue in pediatric dentistry” meets an interesting issue that is relevant for a specific aspect in dental therapy. The review is aimed on the discussion of the “classification of caries and the use of biomaterials in rebuilding dental hard tissues”.
Unfortunately, it is the impression of this reviewer that both topics are only superficially dealt with, if at all, and the title of the manuscript is misleading. It sounds promising but the thematic development is poorly related to the content of the document.
For instance, it remains obscure in this review how materials for caries infiltration, fissure sealants, glass ionomer cements, composite materials or stainless steel crowns are qualified as “biomaterials in rebuilding dental hard tissues”. They might be used in routine caries treatment but not for rebuilding hard tissues. It is amazing that a short comment on the questionable biocompatibility of the infiltrating compound that might be in direct contact to odontoblast processes is missing.
The paragraph on nanohydroxyapatite and it´s positive properties is poorly supported by the reference provided. It appears as of properties weakly supported by experimental data are taken for granted and not cautiously discussed as a promising hypothesis.
Even for materials such as MTA or Biodentin, their mechanisms of action and their suitability as materials for the regeneration of hard dental tissue remains obscure. Likewise, the discussion of dental pulp stem cells is very interesting and relevant per se, this issue is, however, not related to the main topic of this review or the relation to the topic chosen here is not given.
It is also irritating to find some factually misleading statements or paragraphs misplaced. For instance, TEGDMA was considered a material of “high viscosity”, or stem cells are introduced as “non-specialized cells” with “several degrees of their specialization”, and “MTA preparation induces” “interleukin”. What kind of interleukin and how is it effective? It is confusing to find paragraphs about deep carious lesions in a paragraph about fissures sealants, or “endodontic treatment is a consequence of pulp diseases or tooth injuries” in a paragraph about composites.
Reviewer 2 Report
This manuscript concerns the current commercial materials in dental clinic field. Conventional materials are established well. However, the manuscript doesn't express the current state-of-art of dental materials. In particular, recent progress about the use of nanomaterials for dental fields were rarely reported. This manuscript is not appropriate to this journal. Rather, it is recommended to be published in dental field journal.
Reviewer 3 Report
In the manuscript entitled: “Comparison of the application of selected biomaterials promoting the regeneration of hard dental tissue in pediatric dentistry”, the authors identified regenerative dentistry based on understanding both the biological processes of healing and the different stages of tooth development.
The authors found that dental tissue regeneration technology is such a challenge. This scientific research focuses on pediatric dentistry and its classification of caries and the use of biomaterials in rebuilding dental hard tissues.
The authors concluded that there are several methods described, including classical conservative methods like caries infiltration or stainless-steel crowns, as well as the still-developing unique methods involving the use of nanohydroxyapatite and stem cells.
Major comments:
In general, the idea and innovation of this study, regards analysis of nanohydroxyapatite and stem cells in dentistry is interesting, because the role these factors in dentistry are validated but further studies on this topic could be an innovative issue in this field could be open a creative matter of debate in literature by adding new information. Moreover, there are few reports in the literature that studied this interesting topic with this kind of study design.
The study was well conducted by the authors; However, there are some concerns to revise that are described below.
The introduction section resumes the existing knowledge regarding the important factor linked with inflammatory mediators, nanohydroxyapatite and stem cells.
However, as the importance of the topic, the reviewer strongly recommends, before a further re-evaluation of the manuscript, to update the literature through read, discuss and must cites in the references with great attention all of those recent interesting articles, that helps the authors to better introduce and discuss the role of biomarkers released during inflammatory conditions (Galectin, NLRP3): 1) Isola G, Polizzi A, Alibrandi A, Williams RC, Lo Giudice A. Analysis of galectin-3 levels as a source of coronary heart disease risk during periodontitis. J Periodontal Res. 2021 Feb 28. doi: 10.1111/jre.12860. 2) Isola G, Polizzi A, Santonocito S, Alibrandi A, Williams RC. Periodontitis activates the NLRP3 inflammasome in serum and saliva. J Periodontol. 2021 May 19. doi: 10.1002/JPER.21-0049.
The authors should be better specified, at the end of the introduction section, the rational of the study and the aim of the study. In the material and methods section, should better clarify the statistical software used. Moreover, please more extend the results on stem cells in juvenile periodontitis
The discussion section appears well organized with the relevant paper that support the conclusions, even if the authors should better discuss the relationship between early biomarkers for the detection of inflammatory diseases. The conclusion should reinforce in light of the discussions.
In conclusion, I am sure that the authors are fine clinicians who achieve very nice results with their adopted protocol. However, this study, in my view does not in its current form satisfy a very high scientific requirement for publication in this journal and requests a revision before a futher re-evaluation of the manuscript.
Minor Comments:
Abstract:
- Better formulate the abstract section by better describing the aim of the study
Introduction:
- Please refer to major comments
- Page 12 last paragraph: Please reorganize this paragraph that is not clear